# Aquilariae Lignum Methylene Chloride Fraction Attenuates IL-1β-Driven Neuroinflammation in BV2 Microglial Cells

**DOI:** 10.3390/ijms21155465

**Published:** 2020-07-30

**Authors:** Jin-Seok Lee, Yoo-Jin Jeon, Ji-Yun Kang, Sam-Keun Lee, Hwa-Dong Lee, Chang-Gue Son

**Affiliations:** 1Institute of Bioscience & Integrative Medicine, Dunsan Hospital of Daejeon University, Daejeon 35235, Korea; neptune@dju.ac.kr (J.-S.L.); jyj940916@naver.com (Y.-J.J.); kangjy0118@naver.com (J.-Y.K.); 2Department of Applied Chemistry, Oriental Medicine Collage of Daejeon University, Daejeon 35235, Korea; lsk236@dju.kr; 3National Institute for Korean Medicine, 94, Hwarang-ro, Gyeongsan-si, Gyeongsangbuk-do 38540, Korea; herb@nikom.or.kr

**Keywords:** Aquilariae Lignum, neuroinflammation, microglia, NLRP3 inflammasome

## Abstract

Microglial hyperactivation and neuroinflammation are known to induce neuronal death, which is one of the main causes of neurodegenerative disorders. We previously found that Aquilariae Lignum extract attenuated both neuronal excitotoxicity and neuroinflammation in vivo and in vitro. For further analysis, we extracted the methylene chloride fraction of Aquilariae Lignum to determine the bioactive compounds. In this study, we investigated the anti-neuroinflammatory effects and underlying mechanisms of the Aquilariae Lignum fraction (ALF) using lipopolysaccharide (LPS)-stimulated BV2 microglial cells. BV2 cells were pretreated with ALF (0.5, 1, and 2.5 μg/mL) before treatment with LPS (1 μg/mL). Pretreatment with ALF significantly attenuated the LPS-induced overproductions of nitric oxide (NO), cyclooxygenase-2 (COX-2), prostaglandin E_2_ (PGE_2_), and interleukin (IL)-1β. These anti-inflammatory effects were supported by ALF-mediated modulation of the nuclear factor-kappa B (NF-κB) pathway. Furthermore, ALF exerted strong anti-inflammasome effects, as shown by IL-1β-specific inhibitory activity, but not activity against tumor necrosis factor (TNF)-α, along with inhibition of caspase-1 activity and NACHT, LRR, and PYD domain-containing protein 3 (NLRP3)-related molecules. These results indicate the potent anti-neuroinflammatory activity of ALF and that its underlying mechanism may involve the regulation of NLRP3 inflammasome-derived neuroinflammation in microglial cells.

## 1. Introduction

The inflammatory response is an immune defense mechanism to cope with harmful environmental conditions [1]. Microglia, innate immune cells in the brain, play important roles in maintenance of the physiologic status and the pathogenesis of neurodegenerative diseases [2]. In particular, the reactive microglial phenotype can lead to neuronal damage via excessive release of inflammatory cytokines, such as tumor necrosis factor (TNF)-α or interleukin (IL)-1β [3,4]. Many clinical studies have reported the elevation of microglial-derived neuroinflammation in patients with neurodegenerative diseases, including Alzheimer’s disease [5,6].

Although various mechanisms are known to contribute to neuroinflammation, the NACHT, LRR, and PYD domain-containing protein 3 (NLRP3) inflammasome pathway is emerging as an important pathologic process [7]. Microglial NLRP3 inflammasome activation mediates the release and maturation of IL-1β, which is dependent on caspase-1 activity. In a study by Sarsella et al., patients with Alzheimer’s disease had higher mRNA expression of caspase-1 (5-fold) and NLRP3 (7-fold) than healthy controls [8]. Furthermore, previous animal studies have shown that microglial NLRP3-derived IL-1β induces synaptic loss, neuronal apoptosis, and even structural remodeling in the hippocampus [9,10]. In an Alzheimer’s disease mouse model, caspase-1 inhibitor VX-765 dose-dependently reversed cognitive impairment and neuropathology [11]. Accordingly, targeting the NLRP3 inflammasome pathway is considered a therapeutic strategy for neurodegenerative diseases [12,13].

Aquilariae Lignum, also called agarwood, has been used for treating anxiety, pain or inflammation in Southeast and East Asian regions [14]. We previously reported that Aquilariae Lignum has the potential to inhibit microglial overactivation in the mouse hippocampus [15]. In addition, we found neuroprotective effects of Aquilariae Lignum against glutamate-induced excitotoxicity in HT22 cells [16]. The above findings suggest the involvement of microglia-derived neuroinflammation as a target mechanism of Aquilariae Lignum. During the process of identifying the bioactive compounds, we further found that the methylene chloride fraction of Aquilariae Lignum has effective activity at a low dose.

To explore the direct mechanisms underlying the activity of Aquilariae Lignum, we investigated its pharmacological actions on microglia-mediated neuroinflammation, especially the NLRP3 inflammasome pathway, using BV2 murine microglial cells.

## 2. Results

### 2.1. Fingerprinting of ALF

A total of five major peaks were detected at retention times of A (8.13 min), B (13.80 min), C (17.70 min), D (20.92 min), and E (28.52 min) at 280 nm (Figure 1A). The quantitative analysis for each peak was conducted relatively (Figure 1B).

### 2.2. Effects on Cell Viability and Nitric Oxide (NO) Production

A concentration of Aquilariae Lignum fraction (ALF) up to 2.5 μg/mL did not affect the viability of the BV2 cells (Figure 2A). Lipopolysaccharide (LPS) treatment significantly increased the production of NO by 2.3-fold compared with that of the vehicle-treated cells (*p* < 0.01), whereas pretreatment with ALF completely attenuated these elevations compared with LPS treatment (*p* < 0.01 for all doses, Figure 2B). These effects were also shown by following pretreatment with *N*-acetylcysteine (NAC).

### 2.3. Effects on the Molecules Associated with Inflammation

The level of prostaglandin E_2_ (PGE_2_) was significantly increased by treatment with LPS (2.1-fold, *p* < 0.01), whereas pretreatment with ALF dose-dependently inhibited the level of this molecule (*p* < 0.01 for all doses, Figure 3A) compared with LPS treatment. Moreover, pretreatment with ALF substantially attenuated increased expression of inducible nitric oxide synthase (iNOS) and cyclooxygenase-2 (COX-2) at both the mRNA and protein levels following exposure to LPS (*p* < 0.05 or *p* < 0.01, Figure 3B–D). NAC had similar effects as ALF.

### 2.4. Effects on Nuclear Factor-Kappa B (NF-κB) Translocation

LPS significantly induced NF-κB translocation into the nucleus, as shown by the immunofluorescence results and nucleic expression of p65 (*p* < 0.01), while pretreatment with ALF notably attenuated the NF-κB signals as well as the protein expression of nucleic p65 (*p* < 0.05 or *p* < 0.01, Figure 4A–D) compared with LPS treatment. Pretreatment with NAC similarly prevented these effects.

### 2.5. Effects on Inflammatory Cytokines

LPS treatment significantly increased the serum levels of TNF-α (10.6-fold, *p* < 0.01) and IL-1β (1.6-fold, *p* < 0.01), while 2.5 μg/mL of ALF exhibited a notable inhibitory effect on IL-1β (*p* < 0.05, Figure 5A,B) compared with LPS treatment. These results corresponded to gene expression, as shown by inhibition of the IL-1β mRNA levels (*p* < 0.05 or *p* < 0.01, Figure 5C). In contrast, pretreatment with ALF increased TNF-α at both the protein and gene levels compared to that in the LPS-treated cells (*p* < 0.05, Figure 5A,C). NAC had no effect on either cytokine.

### 2.6. Effects on the NLRP3 Inflammasome Pathway

LPS significantly increased the protein levels of NLRP3 inflammasome components, including NLRP3, ASC, and mature-IL-1β (*p* < 0.01 for all proteins), and caspase-1 activity was significantly increased by 1.8-fold (*p* < 0.01) compared with that in the vehicle-treated cell. Meanwhile, these elevations were substantially normalized by pretreatment with ALF (*p* < 0.05 or *p* < 0.01, Figure 6A–C). NAC similarly inhibited activation of the NLRP3 inflammasome.

## 3. Discussion

Neuroinflammation contributes to the pathology of neurodegenerative disorders, such as Alzheimer’s and Parkinson’s disease, as well as neuropsychiatric diseases, including major depressive disorders [17,18]. Under neuroinflammation, excessive release of inflammatory cytokines could alter the vascular remodeling of the blood–brain barrier (BBB), which eventually leads to neuronal death [19,20]. Our previous studies showed that ethanol extract of Aquilariae Lignum attenuates oxidative and inflammatory alterations in the mouse hippocampus and prevents hippocampal excitotoxicity [15,16].

To elucidate the underlying mechanism of Aquilariae Lignum against neuroinflammatory conditions, we adopted LPS-stimulated BV2 microglial cells. This LPS-induced model of neuroinflammation using murine BV2 microglial cells has been commonly used to investigate pharmacological effects [21]. As expected, even at the lowest concentration, pretreatment with ALF completely inhibited the production of both NO and PGE_2_ without affecting cell viability (Figure 2A,B and Figure 3A). This inhibitory activity was confirmed by both the protein and gene expression of iNOS and COX-2 (Figure 3B,D). Products of neuroinflammation, such as NO and PGE_2_, are highly generated from microglial cells, which are frequently observed in the hippocampus of patients with Alzheimer’s disease [22]. Numerous studies have shown that nonsteroidal anti-inflammatory drugs (NSAIDs), which are COX-2 inhibitors, reduce the risk of developing Alzheimer’s disease and enhance learning and memory [23,24]. Moreover, an increase in microglial COX-2 expression was shown in patients with Parkinson’s disease [25].

These inflammatory byproducts are mainly primed by NF-κB as a key activator of inflammation [26]. Previously, NF-κB activity was shown to be 70-fold higher in the dopaminergic neurons of Parkinson’s patients than in those of healthy controls [27]. Blockade of NF-κB transcription inhibits the release of proinflammatory cytokines, including TNF-α and IL-1β. Therefore, in the brain, the overproduction of proinflammatory cytokines could be considered a pathological hallmark of neurodegenerative diseases [28,29]. The modulation of NF-κB is known to have strong potential for the treatment of neurodegenerative disorders, and several influential natural candidates such as curcumin and (−)-epigallocatechin-3-gallate have been suggested in clinical practice [30,31]. In the present study, ALF treatment blocked NF-κB translocation into the nucleus (Figure 4A–D) and consequently attenuated the expression of IL-1β at both the protein and gene levels, although TNF-α was not affected (Figure 5A–C). Our observations were consistent with previous reports showing that 6-amino-4-(4-phenoxyphenylethylamino) quinazoline, an NF-κB inhibitor, reduced the release of IL-1β in radiation-induced BV2 microglial cells without any effects on TNF-α production [32]. These phenomena are also often associated with NLRP3 inflammasome inhibitors, such as MCC950 [33,34]. Accordingly, we further explored the underlying mechanism of ALF activity by focusing on the NLRP3 inflammasome pathway.

The NLRP3 inflammasome complex triggers caspase-1 activation, which is one of the causes of the pathogenesis of neurodegenerative diseases [11,35]. Intriguingly, our results indicated that ALF substantially attenuated the activation of NLRP3 inflammasome components, such as NLRP3 and ASC (Figure 6A,C). Consistent with this finding, mature IL-1β and caspase-1 activity were significantly inhibited by pretreatment with ALF (Figure 6B,C). One study revealed that Z-YVAD-fmk, a caspase-1 inhibitor, prevented human neuronal cell death and caspase-6 activation [36]. Caspase-1 plays an important role in the maturation of IL-1β from pro-IL-1β, and its hyperactivity contributes to programmed neuronal death and microglial activation [37]. High levels of glutamate released by microglial activation induce excitotoxic neuronal death [38]. Accordingly, we evaluated the neuroprotective potential of ALF against glutamate-induced excitotoxicity. As expected, 2.5 μg/mL ALF notably reversed the neuro-excitotoxicity of HT22 hippocampal neuronal cells, was nontoxic (Appendix A), and had effects that were superior to those of the positive control.

On the other hand, NO was recently shown to inhibit NLRP3 inflammasome activity via thiol nitrosylation in human macrophages [39,40]. Regarding the interaction between PGE2 and NLRP3 inflammasome, conflicting data were reported. PGE2 induced the inactivation of NLRP3 inflammasome by modulation of EP4 receptor and intracellular cAMP in human primary monocyte-derived macrophages [41], while it activated NLRP3 inflammasome in human retinal microvascular endothelial cells and THP-1 human macrophages [42,43]. In our study, however, ALF was found to suppress NO, PGE2, and NLRP3 inflammasome. We need further investigations to figure out whether our ALF-derived results are specific to brain microglia by comparing to neuronal cells and astrocytes.

A few putative active compounds of Aquilariae Lignum, such as buagafuran and *N*-trans-feruloyltyramine, have previously been shown to have pharmacological effects on neuronal cell death [44,45]. Furthermore, several sesquiterpenes from resinous wood (*Aquilaria sinensis*) exhibited anti-neuroinflammatory effects in BV2 microglial cells [46]. Previously, we also proved the neuroprotective activity of Aquilariae Lignum ethanol extracts on hippocampal excitotoxicity (in vitro), microglial activation (in vitro and in vivo), and hippocampal oxidative damage (in vivo). However, those studies did not manage to explain all underlying mechanisms and did not consider NLRP3 inflammasome and related mediators in particular.

Thus, we herein suggest that the methylene chloride fraction of Aquilariae Lignum is a strong candidate for alleviating the microglia-derived neuroinflammatory response, and its underlying mechanism may involve the regulation of the NLRP3 inflammasome pathway. However, further investigations are required for the identification of corresponding compounds and characterization of actions in other brain cells, including neurons and astrocytes.

## 4. Materials and Methods

### 4.1. Materials

The following reagents were obtained from Sigma-Aldrich (St. Louis, MO, USA): L-glutamic acid monosodium salt monohydrate, *N*-acetylcysteine (NAC), Tween 20, 4′,6-diamidino-2-phenylindole dihydrochloride (DAPI), aqueous mounting buffer, sodium hydroxide, sulfanilamide, *N*-(1-naphthyl)-ethylenediamine dihydrochloride, and phosphoric acid. Other reagents were obtained from the following manufacturers: Acetyl alcohol, paraformaldehyde, H_2_O_2_, citrate buffer, methylene alcohol and Triton X-100 were obtained from Duksan Science, Seoul, Korea; Daejung Chemicals & Metals Co., Siheung, Korea; and Junsei Chemical Co., Ltd., Tokyo, Japan. Fetal bovine serum (FBS), Dulbecco’s modified Eagle’s medium (DMEM), Dulbecco’s phosphate-buffered saline (DPBS), penicillin–streptomycin solution, and trypsin–ethylenediaminetetraacetic acid (EDTA) solution were obtained from Welgene, Daegu, Korea; bovine serum albumin (BSA) was obtained from GenDEPOT, Barker, TX, USA; and *n*-butanol was obtained from J.T. Baker, Mexico City, Mexico. Normal goat serum and primary and secondary antibodies were obtained from Abcam, Cambridge, MA, USA; Thermo-Fisher Scientific, Carlsbad, CA, USA; Santa Cruz Biotechnology, Santa Cruz, CA, USA; and AMRESCO, Solon, OH, USA.

### 4.2. Preparation of the Aquilariae Lignum Methylene Chloride Fraction (ALF)

Resinous heartwood of agarwood (Aquilariae Lignum) was purchased from an herbal pharmaceutical company (Dae Han Bio Pharm Inc., Gyeonggi-do, Korea). Methylene chloride extracts were prepared as follows: Five grams of Aquilariae Lignum powder were mixed with 200 g of methylene chloride and stirred for 24 h at room temperature. After filtration of the supernatants using a 0.2 µm nylon filter, the supernatant fluid was washed two times with 100 mL of deionized water, and the methylene chloride layer was then dried with sodium sulfate. The removal of volatiles by a rotary evaporator gave 0.62 g of an orange gummy material.

### 4.3. Fingerprinting Analysis of ALF

To identify the chemical composition of ALF, we conducted high-performance liquid chromatography (HPLC). Briefly, 5 mg of ALF was dissolved in 1 mL of 90% methanol, and the solutions were filtered (0.45 μm). Then, the filtrate was subjected to UHPLC-MS using an LTQ Orbitrap XL linear ion trap MS system (Thermo Scientific Co., San Jose, CA, USA) equipped with an electrospray ionization source. Separation was performed on an Accela HPLC system using an Acquity BEH C18 column (1.7 μm, 2.1 × 150 mm; Waters, Milford, MA, USA). The mobile phase conditions were prepared as follows: (A) distilled water and (B) acetonitrile, both of which contained 0.1% formic acid. The column was eluted at a flow rate of 0.4 mL/min with the following gradients: 0–1 min, 5% B (isocratic); 1–20 min, 5–70% B (linear gradient); 20–24 min, 70–100% B (linear gradient); and 24–27 min, 100% B (isocratic). The photodiode array detector was set to measure a range of 200–700 nm.

### 4.4. Cell Culture and Cell Viability

Murine microglial cells (BV2) were cultured in DMEM supplemented with 10% FBS and 1% penicillin–streptomycin. BV2 cells were incubated at 37 °C under 5% CO_2_, and the cells (2 × 10^4^ cells/well) were seeded into 96-well microplates and then incubated for 12 h. Then, the cells were pretreated with ALF (0.5, 1, and 2.5 μg/mL) or a positive control (NAC, 100 μM) for 24 h. To evaluate the cytotoxicity of ALF, we determined the cell viability with a WST-8 assay (EZ-Cytox, DoGen, Korea). Absorbance at 450 nm was measured using a UV spectrophotometer (Molecular Devices, Sunnyvale, CA, USA).

### 4.5. NO Assay

BV2 cells were seeded at 2 × 10^4^ cells/well into 96-well microplates. After incubation for 12 h, the cells were pretreated with ALF (0.5, 1, and 2.5 μg/mL) or positive control (NAC, 100 μM) for 2 h before exposure to 1 μg/mL LPS. After incubation for 24 h, the supernatants were mixed with an equal volume of Griess reagent (1% sulfanilamide/0.1% *N*-(1-naphthyl)-ethylenediamine dihydrochloride/2.5% H_3_PO_4_). After incubation for 15 min at 37 °C, absorbance at 405 nm was measured using a UV spectrophotometer (Molecular Devices).

### 4.6. PGE_2_ Assay

BV2 cells (2 × 10^4^ cells/well) were seeded into 60 mm dishes, and the experimental conditions were the same as those for the NO assay. The PGE_2_ concentration of the supernatants was determined using an ELISA kit (R&D Systems, Minneapolis, MN, USA). The absorbance at 450 nm was measured using a UV spectrophotometer (Molecular Devices).

### 4.7. Proinflammatory Cytokine Activity

Under the same cell culture conditions as the PGE_2_ assay, the proinflammatory cytokine levels of the supernatants were determined using a commercially available enzyme immunoassay (EIA) kit for TNF-α (BD Biosciences, San jose, CA, USA) and IL-1β (R&D Systems Inc., Minneapolis, MN, USA). The absorbance was read at 450 nm using a UV spectrophotometer (Molecular Devices).

### 4.8. Caspase-1 Assay

Under the same cell culture conditions as the PGE_2_ assay, cells were prepared in radioimmunoprecipitation assay (RIPA) lysis buffer. Caspase-1 activity of the lysates was determined using a mouse caspase-1 ELISA kit (Novus Biologicals, Littleton, CO, USA) according to the manufacturer’s protocol. The absorbance was read at 450 nm using a UV spectrophotometer (Molecular Devices).

### 4.9. Western Blot Analysis

Under the same cell culture conditions as the caspase-1 assay, cell lysates were separated by polyacrylamide gel electrophoresis and transferred to polyvinylidene fluoride (PVDF) membranes. After the membranes were blocked in 3% BSA for 1 h, they were probed with primary antibodies against iNOS (1:200, PA1-036), COX-2 (1:1000, sc-514489), phospho-NF-κB p65 (1:2000, ab86299), NF-κB p65 (1:1000, sc-372), lamin B1 (1:100, sc-374015), NLRP3 (1:1000, ab2141185), ASC (1:200, sc-271054), IL-1β (1:2500, ab9722), and β-actin (1:1000, MA5-11869) overnight at 4 °C. The membranes were washed and incubated with HRP-conjugated anti-mouse (1:2500, against COX-2, ASC, Lamin B1, and β-actin) or anti-rabbit (1:2500, against iNOS, phospho-NF-κB p65, NF-κB p65, NLRP3, and IL-1β) antibodies for 45 min. The protein was visualized using an enhanced chemiluminescence (ECL) advanced kit (Thermo Fisher Scientific, Carlsbad, CA, USA). Protein expression was observed using the FUSION Solo System (Vilber Lourmat, Collegien, France). The intensity was semiquantified using ImageJ 1.46 software (NIH, Bethesda, MD, USA).

For determination of the nuclear and cytoplasmic expression of NF-κB p65, the cytosolic and nucleic extracts of BV2 cells were separated using NE-PER Nuclear and Cytoplasmic Extraction Reagents (Thermo Fisher Scientific, CA, USA) according to the manufacturer’s instructions.

### 4.10. RNA Isolation and Quantitative Real-Time PCR

BV2 cells were seeded at 2 × 10^4^ cells/well into 60 mm dishes. After incubation for 12 h, the cells were pretreated with ALF (0.5, 1, and 2.5 μg/mL) or the positive control (NAC, 100 μM) for 2 h before exposure to 1 μg/mL LPS. After incubation for 6 h, total RNA was isolated using an RNeasy Mini kit (Qiagen; Valencia, CA, USA), and cDNA was synthesized using a High-Capacity cDNA reverse transcription kit (Ambion, Austin, TX, USA). Real-time PCR was performed using SYBR Green PCR Master Mix (Applied Biosystems, Carlsbad, CA, USA), and PCR amplification was performed using a standard protocol with Rota-Gene Q real-time PCR (Qiagen; Valencia, CA, USA). Information regarding the primer sequences is summarized in Table 1.

### 4.11. Immunofluorescence Staining

BV2 cells were seeded at 2 × 10^4^ cells/well into 60 mm dishes. After incubation for 12 h, the cells were pretreated with ALF (0.5, 1, and 2.5 μg/mL) or the positive control (NAC, 100 μM) for 2 h before exposure to 1 μg/mL LPS. After incubation for 1 h, the cells were sequentially washed with PBST (0.05% Tween 20 in PBS), fixed with 4% paraformaldehyde for 20 min, permeabilized with 0.1% Triton X-100 for 10 min, and blocked with 1% BSA in PBST for 30 min. The cells were incubated with phospho-NF-κB p65 antibody (1:100) for 1 h at room temperature. After washing, the cells were incubated with Alexa Fluor 488-conjugated secondary antibody for 1 h in the dark. The nuclei were counterstained with DAPI solution. Fluorescent images were observed under an Axio-phot microscope (Carl Zeiss, Jena, Germany).

### 4.12. Statistical Analysis

All data are expressed as the mean ± standard deviation (SD). The statistically significant differences between the groups were evaluated by one-way analysis of variance (ANOVA) followed by post hoc multiple comparisons with Tukey’s HSD test using IBM SPSS statistics software, ver. 20.0 (SPSS Inc., Chicago, IL, USA). Differences at *p* < 0.05 or *p* < 0.01 were considered statistically significant.

## Figures and Tables

**Figure 1 ijms-21-05465-f001:**
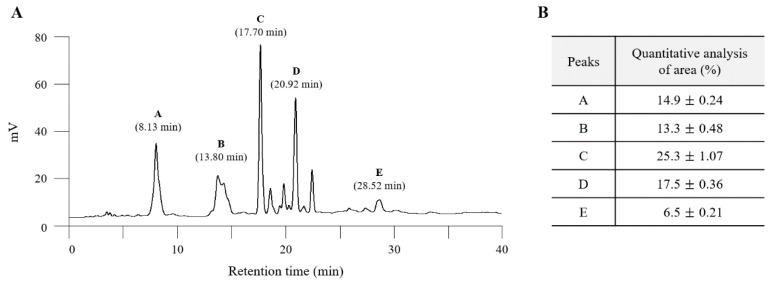
Fingerprinting analysis of the Aquilariae Lignum fraction (ALF). Five peaks were detected at each retention time (**A**), and their areas were semiquantified (**B**). ALF was subjected to UHPLC-MS, and a chromatogram was obtained at a wavelength of 280 nm.

**Figure 2 ijms-21-05465-f002:**
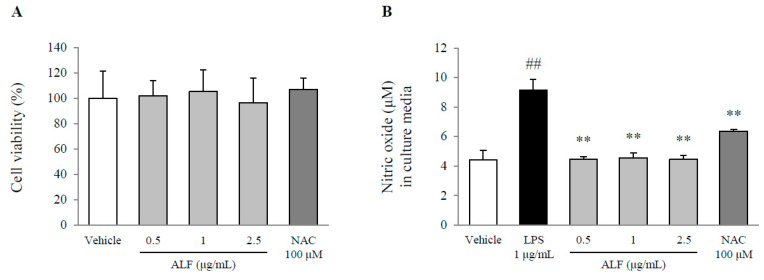
Cell viability and nitric oxide levels. BV2 cells were pretreated with ALF for 2 h before exposure to lipopolysaccharide (LPS) (1 μg/mL) for 24 h. The cytotoxicity of ALF (**A**) and the levels of nitric oxide (**B**) in BV2 cells were determined. The data are expressed as the mean ± SD (*n* = 6). ## *p* < 0.01 compared with the vehicle-treated cells; ** *p* < 0.01 compared with the LPS-treated BV2 cells.

**Figure 3 ijms-21-05465-f003:**
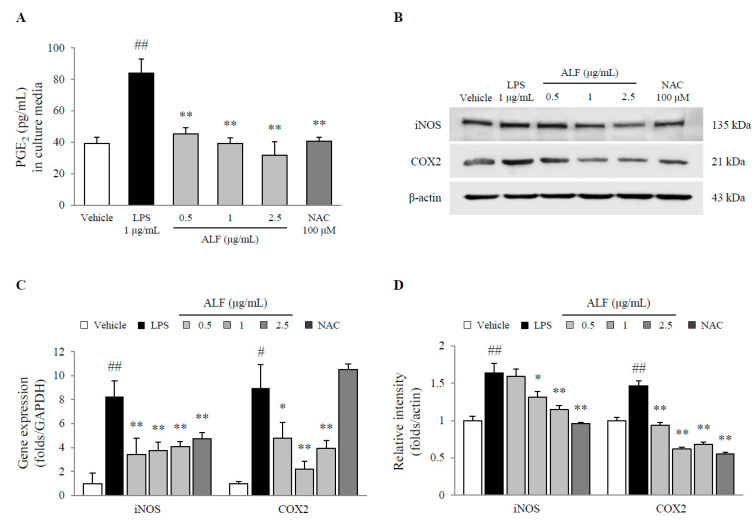
Prostaglandin E_2_ (PGE_2_) production and inducible nitric oxide synthase (iNOS) and cyclooxygenase-2 (COX-2) expression in BV2 cells. Cell supernatant PGE_2_ levels (**A**), protein expression of iNOS and COX-2 (**B**), and gene expression of iNOS and COX-2 relative to that of GAPDH (**C**) were determined using ELISA (*n* = 6), Western blot (*n* = 3), or real-time PCR (*n* = 3) methods. Semiquantification of protein expression was performed with normalization to β-actin (**D**). The data are expressed as the mean ± SD. # *p* < 0.05 and ## *p* < 0.01 compared with the vehicle-treated cells; * *p* < 0.05 and ** *p* < 0.01 compared with the LPS-treated BV2 cells.

**Figure 4 ijms-21-05465-f004:**
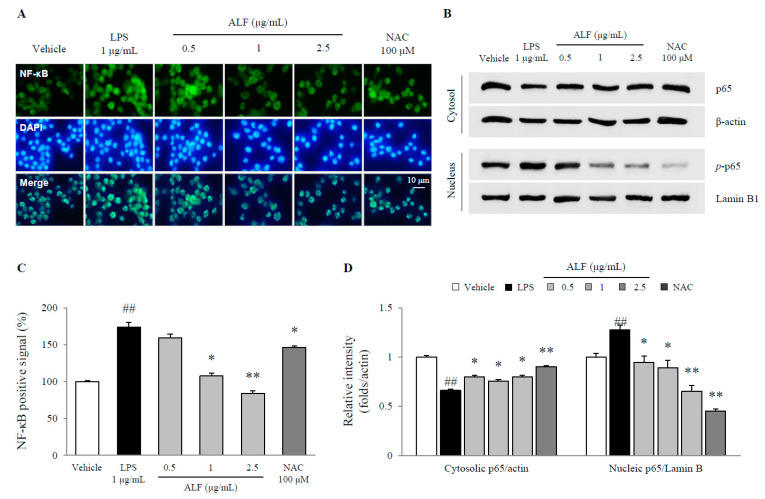
Nuclear factor-kappa B (NF-κB) signals in BV2 cells. Signals for nuclear translocation of NF-κB (**A**) and cytosolic and nucleic p65 expression (**B**) were determined using IF staining or Western blot methods. The intensity of the histological signals (**C**) and semiquantification of protein expression relative to β-actin or lamin B1 (**D**) were analyzed. The data are expressed as the mean ± SD (*n* = 3). ## *p* < 0.01 compared with the vehicle-treated cells; * *p* < 0.05 and ** *p* < 0.01 compared with the LPS-treated BV2 cells.

**Figure 5 ijms-21-05465-f005:**
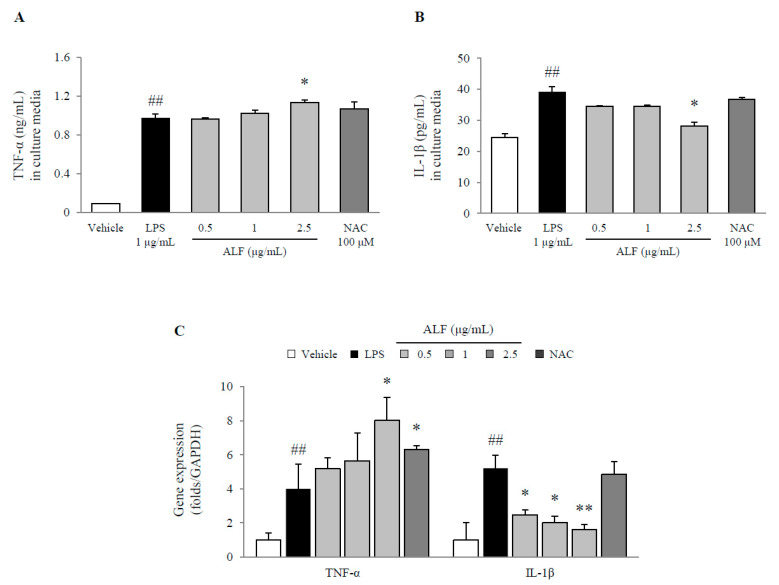
Tumor necrosis factor (TNF)-α and interleukin (IL)-1β levels in BV2 cells. Proinflammatory cytokines TNF-α (**A**) and IL-1β (**B**) in the cell supernatant and gene expression relative to GAPDH (**C**) in the cell lysate were determined using ELISA (*n* = 6) and real-time PCR (*n* = 3), respectively. The data are expressed as the mean ± SD. ## *p* < 0.01 compared with the vehicle-treated cells; * *p* < 0.05 and ** *p* < 0.01 compared with the LPS-treated BV2 cells.

**Figure 6 ijms-21-05465-f006:**
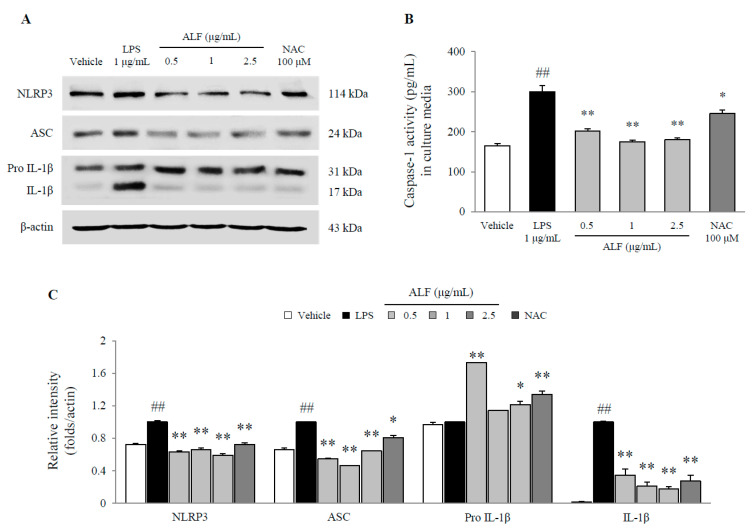
NACHT, LRR, and PYD domain-containing protein 3 (NLRP3) inflammasome pathway in BV2 cells. Protein expression of NLRP3, ASC, and pro and mature-IL-1β was determined using Western blot (**A**). Caspase-1 activity in cell lysates was determined using ELISA (**B**). Semiquantification of the protein expression was performed with normalization to β-actin (**C**). The data are expressed as the mean ± SD (*n* = 3). ## *p* < 0.01 compared with the vehicle-treated cells; * *p* < 0.05 and ** *p* < 0.01 compared with the LPS-treated BV2 cells.

**Table 1 ijms-21-05465-t001:** Gene sequence summary.

Gene	Primer	Primer Sequence (5′→ 3′)
iNOS	Forward	GGC AGC CTG TGA GAC CTT TG TGC ATT GGA AGT GAA GCG TTT
	Reverse	
COX-2	Forward	CAG CAA CTC CTT GCT GTT CC TGG GCA AAG AAT GCA AAC ATC
	Reverse	
TNF-α	Forward	CTC CCA GGT TCT CTT CAA GG TGG AAG ACT CCT CCC AGG TA
	Reverse	
IL-1β	Forward	AAG TTG ACG GAC CCC AAA AGA TTG ATG TGC TGC GAG AT
	Reverse	
GAPDH	Forward	CAT GGC CTT CCG TGT TCC T CCT GCT TCA CCA CCT TCT TGA
	Reverse	

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
