# Peer review of "Aquilariae Lignum Methylene Chloride Fraction Attenuates IL-1β-Driven Neuroinflammation in BV2 Microglial Cells"

_ijms, 2020, doi:10.3390/ijms21155465_

Round 1

Reviewer 1 Report

Previous studies by the authors have shown that the ethanol extract of Aquilariae Lignum attenuates oxidative and inflammatory alterations in the mouse hippocampus and prevents hippocampal excitotoxicity. The authors in these in vitro studies assess the anti-neuroinflammatory effects and underlying mechanisms of the Aquilariae Lignum methylene chloride fraction (ALF) using lipopolysaccharide (LPS)-stimulated BV2 microglial cells. The BV2 cells were pretreated with ALF (0.5, 1, 2.5 μg/mL) before treatment with LPS (1 μg/mL). Pretreatment with ALF significantly attenuated the LPS-induced overproductions of nitric oxide (NO), cyclooxygenase-2 (COX-2), prostaglandin E2 (PGE2), and interleukin (IL)-1β. ALF-mediated modulation of the nuclear factor-kappa B (NF-κB) pathway was found. ALF had IL-1β-specific inhibitory activity with inhibition of caspase-1 activity and NACHT, LRR and PYD domain-containing protein 3 (NLRP3)-related molecules. The authors indicate that the potent anti-neuroinflammatory activity of ALF may primarily involve the regulation of NLRP3 inflammasome-derived neuroinflammation in microglial cells.

Comments:

1.      The methylene chloride fraction of Aquilariae Lignum and its regulation of the NLRP3 inflammasome pathway as the underlying mechanism with relevance to the prevention of microglial hyperactivation and neuroinflammation that are known to induce neuronal death in neurodegenerative disorders is of major interest to brain research.

2.      The role of nitric oxide (NO) as a critical negative regulator of the NLRP3 inflammasome has been reported in biology. Nitric oxide regulates NF-κB activation, having been reported to either stimulate or inhibit activity depending on the cell-type. NF-κB regulates COX-2 gene expression.

3.      COX-2 regulates the activation of the NLRP3 inflammasome derived IL-1β Production    and cell biology experiments indicate that it may be a new potential therapeutic target in NLRP3 inflammasome-related diseases. COX-2 catalyzes the synthesis of prostaglandin E2 and increases IL-1β secretion in cell experiments.

4.      In cell studies PGE2 inhibits NLRP3 Inflammasome Activation as a regulator of inflammation.

Questions:

1.      Do the results indicate that ALF may primarily involve the regulation of NLRP3 inflammasome-derived neuroinflammation in microglial cells and the effects of NO, COX-2 and PGE2 are secondary?

2.      Are the interactions of NO, COX-2 , PGE-2 and NLRP3 inflammasome-IL-1β production different in microglial cells with relevance to other biological cells with relevance to the authors results with ALF?

3.       

Comment:

5.      The authors could include a schematic figure in this manuscript to highlight the role of ALF on anti-neuroinflammatory activity with the primary NLRP3 inflammasome pathway in microglial cells.

RELEVANT REFERENCS:

1.      Eduardo Hernandez-Cuellar, Kohsuke Tsuchiya, Hideki Hara, Rendong Fang, Shunsuke Sakai, Ikuo Kawamura, Shizuo Akira and Masao Mitsuyama. Cutting Edge: Nitric Oxide Inhibits the NLRP3 Inflammasome. J Immunol  December 1, 2012,  189  (11)  5113-5117.

2.      Mao K, Chen S, Chen M, et al. Nitric oxide suppresses NLRP3 inflammasome activation and protects against LPS-induced septic shock. Cell Res. 2013;23(2):201-212. doi:10.1038/cr.2013.6

3.      J R Matthews, C H Botting, M Panico, H R Morris, and R T Hay. Inhibition of NF-kappaB DNA binding by nitric oxide. Nucleic Acids Res. 1996 Jun 15; 24(12): 2236–2242.

4.      Mendes AF, Carvalho AP, Caramona MM, Lopes MC. Role of nitric oxide in the activation of NF-kappaB, AP-1 and NOS II expression in articular chondrocytes. Inflamm Res. 2002;51(7):369-375.

5.      Kaltschmidt, B., Linker, R.A., Deng, J. et al. Cyclooxygenase-2 is a neuronal target gene of NF-κB. BMC Molecular Biol 3, 16 (2002).

6.      Hua KF, Chou JC, Ka SM, et al. Cyclooxygenase-2 regulates NLRP3 inflammasome-derived IL-1β production. J Cell Physiol. 2015;230(4):863-874.

7.      Milena Sokolowska, Li-Yuan Chen, Yueqin Liu, Asuncion Martinez-Anton, Hai-Yan Qi , Carolea Logun , Sara Alsaaty , Yong Hwan Park , Daniel L Kastner , Jae Jin Chae, James H Shelhamer.   Prostaglandin E2 Inhibits NLRP3 Inflammasome Activation Through EP4 Receptor and Intracellular Cyclic AMP in Human Macrophages. J Immunol. 2015;194(11):5472-5487.

8.       Zhuang Y. Zhao F. Liang J. Deng X. Zhang Y. Ding G. Zhang A. Jia Z. Huang S. Activation of COX-2/mPGES-1/PGE2 Cascade via NLRP3 Inflammasome Contributes to Albumin-Induced Proximal Tubule Cell Injury. Cell Physiol Biochem 2017;42:797–807

Author Response

Reviewer questions:

  1. Do the results indicate that ALF may primarily involve the regulation of NLRP3 inflammasome-derived neuroinflammation in microglial cells and the effects of NO, COX-2 and PGE2 are secondary?
  • We sincerely appreciate reviewer for the professional comments and helpful questions. As reviewer indicated, NLRP3 inflammasome and inflammatory mediators (NO and/or COX2-mediated PGE2) are known to interact mutually. The present data however could not clarify whether ALF regulate primarily NLRP3 inflammasome or inflammatory mediators (NO and/or COX2-mediated PGE2). We have added those descriptions in ‘Discussion’ section, and will try to find out the issue in the next study.

  1. Are the interactions of NO, COX-2, PGE-2 and NLRP3 inflammasome-IL-1β production different in microglial cells with relevance to other biological cells with relevance to the authors results with ALF?
  • We really appreciate reviewer for the professional questions. Our previous studies showed that Aquilariae Lignum extract acts to relieve neuronal damage and microglial overactivation in mouse hippocampus (Lee et al., 2017) as well as to prevent neuroexcitotoxicity in hippocampal neuronal cell line (Lee et al., 2018). Unfortunately, in those studies, the involvement of NLRP3 inflammasome was not considered. As reviewer mentioned, the role of NO and PGE2 was revealed to inhibit NLRP3 inflammasome activity in phagocytes, especially in macrophage (Eduardo et al., 2012; Milena et al., 2015). These phenomena, however, are still uncertain how to act with other brain cells such as neuron and astrocyte. The further studies are required for the issue. We have explained more about it in ‘Discussion’ section.

  1. The authors could include a schematic figure in this manuscript to highlight the role of ALF on anti-neuroinflammatory activity with the primary NLRP3 inflammasome pathway in microglial cells.
  • We fully agree with reviewer opinion, however, there is a limitation of figure number in this journal guideline. According to reviewer suggestions, the primary role of ALF on NLRP3 inflammasome pathway was reflected in the revised ‘Graphical abstract’.

Reviewer 2 Report

The study by Lee et al entitled “Aquilariae Lignum Methylene Chloride Fraction Attenuates IL-1β-driven Neuroinflammation in BV2 Microglial cells” has been reviewed. The study evaluates the anti-neuroinflammatory effects of the methylene chloride fraction of Aquilariae Lignum using lipopolysaccharide (LPS)-stimulated BV2 microglial cells and the possible involvement of NLRP3 inflammasome pathway.

The manuscript is clear and well designed; nonetheless, the conclusions are not clearly shown by the evidence provided. Minor suggestions can improve the manuscript.

1- Please better characterize the extract identifying the chemical compounds of the main peaks shown in the chromatogram of the methylene chloride fraction (Fig 1) and please quantify at least, the total amount of bioactive compounds present in the extract so that the experimental conditions are reproducible. Clarify this point and provide clear data on this aspect.

2- Better argue the NLRP3 inflammasome pathway in the discussion, accordingly with the biomarkers examined.

So, overall, I am positive about this study, but I would urge the authors to accommodate the recommended changes.

Author Response

1-Please better characterize the extract identifying the chemical compounds of the main peaks shown in the chromatogram of the methylene chloride fraction (Fig 1) and please quantify at least, the total amount of bioactive compounds present in the extract so that the experimental conditions are reproducible. Clarify this point and provide clear data on this aspect.

  • According to reviewer suggestion, we additionally provided the quantitative values of each peak area in ‘Figure 1B’ of the revised manuscript. We have a plan to identify the major active compounds.

2- Better argue the NLRP3 inflammasome pathway in the discussion, accordingly with the biomarkers examined.

  • Thank reviewer for helpful comments. According to reviewer suggestions, we additionally discussed the interaction between NLRP3 inflammasome and other inflammatory mediators (NO and PGE2) in this revised manuscript.

Round 2

Reviewer 1 Report

The authors have ensured that the research is properly verified before being published. The revised manuscript helps to hone in on key points and avoid inadvertent errors. The revised manuscript maintains the high standards for peer reviewed journals.

This manuscript is a resubmission of an earlier submission. The following is a list of the peer review reports and author responses from that submission.

Round 1

Reviewer 1 Report

To assess the direct mechanisms underlying the activity of Aquilariae Lignum the authors investigated its pharmacological actions on microglia-medicated neuroinflammation via the NLRP3 inflammasome pathway using BV2 murine microglial cells. The authors previous studies showed that the ethanol extract of Aquilariae Lignum attenuates oxidative and inflammatory alterations in the mouse hippocampus and prevents hippocampal excitotoxicity. The authors in these experiments assessed and identified the methylene chloride fraction of Aquilariae Lignum at a low dose. LPS-induced model of neuroinflammation using murine BV2 microglial cells was used in these experiments and pretreatment with ALF (even at the lowest concentration) completely inhibited the production of both NO and PGE2 without affecting cell viability. ALF treatment blocked NF-κB translocation into the nucleus and attenuated the expression of IL-1β at both the protein and gene levels. ALF substantially attenuated the activation of NLRP3 inflammasome components, such as NLRP3 and ASC. The authors conclude that methylene chloride fraction of Aquilariae Lignum alleviates microglia-derived neuroinflammatory response via regulation of the NLRP3 inflammasome pathway.

Comments:

  1. The literature shows that the nucleotide-binding oligomerization domain-, leucine-rich repeat- and pyrin domain-containing 3 (NLRP3) inflammasome, a subcellular multiprotein complex that is abundantly expressed in the central nervous system (CNS), can sense and be activated by a wide range of exogenous and endogenous stimuli such as microbes, aggregated and misfolded proteins, and adenosine triphosphate, which results in activation of caspase-1. Activated caspase-1 subsequently leads to the processing of interleukin-1β (IL-1β) and interleukin-18 (IL-18) pro-inflammatory cytokines and mediates rapid cell death. IL-1β and IL-18 drive inflammatory responses through diverse downstream signaling pathways, leading to neuronal damage. Thus, the NLRP3 inflammasome is considered a key contributor to the development of neuroinflammation.
  2. The authors use the methylene chloride fraction that contains more effective bioactive compounds of Aquilariae Lignum when compared to ethanol or methanol extracts.
  3. The LPS-induced model of neuroinflammation was used with murine BV2 microglial cells and pretreatment with ALF completely inhibited the production of both NO and PGE2, attenuated LPS induced NF-κB signals. LPS significantly increased the protein levels of NLRP3 inflammasome components, including NLRP3, ASC, and mature-IL-1β and pretreatment with ALF normalized these levels.

Questions:

Q1. Is the methylene chloride extract of ALF contain bioactive components that bind to LPS in vitro?

Q2. Can the pretreatment of murine BV2 microglial cells with ALF be associated with binding of ALF to cell and nuclear receptors that displace LPS binding interactions?

Q3. Does LPS bind to the nucleotide-binding oligomerization domain-, leucine-rich repeat- and pyrin domain-containing 3 (NLRP3) inflammasome, a subcellular multiprotein complex with ALF pretreatment involved with displacement from this complex?

Q4. Is ALF therapeutics that alleviates microglia-derived neuroinflammatory more efficient than other compounds such as Bushen-Yizhi Formula that inhibits NLRP3 Inflammasome Activation?

RELEVANT REFERENCES:

  1. Limin Song, Lei Pei, Shanglong Yao, Yan Wu, and You Shang. NLRP3 Inflammasome in Neurological Diseases, from Functions to Therapies. Front Cell Neurosci. 2017; 11: 63.
  2. Yousheng Mo et al. Bushen-Yizhi Formula Alleviates Neuroinflammation via Inhibiting NLRP3 Inflammasome Activation in a Mouse Model of Parkinson’s Disease. Volume 2018 |Article ID 3571604 | 12 pages